# Organoids as Tools for Investigating Skin Aging: Mechanisms, Applications, and Insights

**DOI:** 10.3390/biom14111436

**Published:** 2024-11-12

**Authors:** Xin-Yu Wang, Qian-Nan Jia, Jun Li, He-Yi Zheng

**Affiliations:** Peking Union Medical College Hospital (Dongdan Campus), No. 1 Shuaifuyuan Wangfujing Dongcheng District, Beijing 100730, China; wangxinyu@student.pumc.edu.cn (X.-Y.W.); jiaqn@pumch.cn (Q.-N.J.)

**Keywords:** skin aging, aging skin equivalent, in vitro skin models, skin organoid, aging skin organoid

## Abstract

Organoids have emerged as transformative tools in biomedical research, renowned for their ability to replicate the complexity construct of human tissues. Skin aging is a multifaceted biological process, influenced by both intrinsic factors and extrinsic factors. Traditional models for studying skin aging often fall short in capturing the intricate dynamics of human skin. In contrast, skin organoids offer a more physiologically relevant system, reflecting the structural and functional characteristics of native skin. These characteristics make skin organoids highly suitable for studying the mechanisms of skin aging, identifying novel therapeutic targets, and testing anti-aging interventions. Despite their promise, challenges such as limited scalability, reproducibility, and ethical considerations remain. Addressing these hurdles through interdisciplinary research and technological advancements will be essential to maximizing the potential of skin organoids for dermatological research and personalized anti-aging therapies.

## 1. Introduction

Organoids, as three-dimensional (3D) structures derived from stem cells, have become pivotal in biomedical research due to their ability to mimic the complexity and function of human organs [1]. Unlike traditional two-dimensional (2D) cell cultures or animal models, organoids provide a more physiologically relevant environment that mirrors the multicellular organization and functionality of tissues. Therefore, organoids are increasingly being used in biomedical research to study disease mechanisms, tissue development, and therapeutic responses with greater accuracy across various fields, including oncology, regenerative medicine, and genetic disorders [2,3].

Skin aging is a complex biological process influenced by both intrinsic factors [4], such as genetics, and extrinsic factors, including UV exposure and environmental pollutants. These factors collectively contribute to the gradual degradation of skin structure and function, characterized by the breakdown of the extracellular matrix (ECM) [5], decreased cellular proliferation, and heightened oxidative stress [6] in the skin tissues, resulting in the gradual degradation of skin structure and function. The visible manifestations of skin aging, such as wrinkles, loss of elasticity, and hyperpigmentation, result from these underlying molecular and cellular changes. Despite significant progress in our understanding of the biology of skin aging, the animal models and 2D cell cultures often fail to fully capture the complex interactions between the multiple cell types and structural components of human skin over time. These limitations underscore the need for more sophisticated models that can replicate the dynamic processes involved in skin aging [7].

Traditional models have long been used to investigate skin biology. However, 2D cell cultures, animal models, and 3D skin equivalents fall short when it comes to mimicking complex tissue structures and long-term dynamic processes such as chronic inflammation, ECM remodeling, and gradual cellular senescence that occur during skin aging. The limitations of these traditional models underscore the need for more sophisticated and physiologically relevant systems. Under these circumstances, organoid models, particularly skin organoids, have emerged as a promising tool to address the challenges present during the process of skin aging [8,9,10]. By closely replicating the architecture and function of human skin, including critical features such as the dermal–epidermal junction, hair follicles, and sebaceous glands, skin organoids provide a powerful platform for studying the cellular and molecular mechanisms of aging [11,12]. They allow for the investigation of how various factors contribute to the aging process at a level of detail that is difficult to achieve with other models [13].

Exploring the development and application of organoid models in skin aging research is crucial for advancing our understanding of this complex process. This review aims to synthesize the current knowledge on how these innovative models can be utilized to unravel the intricate mechanisms underlying skin aging. By examining recent advancements in organoid technology and their implications for skin biology, this review highlights the potential of organoids to serve as powerful tools in both basic research and therapeutic development. As the field continues to grow, this comprehensive overview will provide valuable insights for researchers looking to harness organoid models to uncover novel therapeutic targets and strategies for combating skin aging.

## 2. A Brief Overview of the Skin

### 2.1. Structure of Skin Tissue

The skin is the largest and one of the most complex organs in the human body [14]. It serves as the body’s first line of defense, and plays vital roles in barrier protection, immune regulation, thermoregulation, and sensory perception. The normal functions of the skin, and the pathological changes that occur with aging, are determined by its unique structure.

The skin consists of three primary layers (Figure 1): the epidermis, dermis, and hypodermis [15]. The epidermis, the outermost layer, is primarily composed of keratinocytes that are continuously renewed through basal stem cell differentiation. This layer also contains melanocytes, responsible for pigmentation and UV protection, and the Langerhans cells, which are crucial for immune responses [16]. Beneath the epidermis lies the dermis, a layer rich in collagen and elastin fibers that provide structural integrity and elasticity. The dermis also houses blood vessels, nerve endings, hair follicles, and sweat glands, which are essential for thermoregulation and sensory functions. The dermal matrix consists of glycosaminoglycans, and supports cellular communication, skin hydration, and resilience. The hypodermis [17], the innermost layer, consists largely of adipose tissue, which insulates the body and cushions underlying structures, while also playing a critical role in energy storage and anchoring the skin to muscles and bones.

### 2.2. Key Signaling Pathways in Skin Tissue Development

The development of skin tissue is intricately linked to evolutionarily conserved signaling pathways, including Wnt/β-catenin, fibroblast growth factors (FGFs), bone morphogenetic proteins (BMPs), and Hedgehog (Hh) signaling [18,19,20]. These pathways guide the formation of skin components during embryogenesis. The Wnt signaling mediates the development of the epidermis from the ectoderm, while the dermis originates from the mesenchyme under the influence of BMPs. FGF signaling, meanwhile, facilitates cellular interactions between the epidermis and dermis, promoting cell proliferation, angiogenesis, and tissue repair. Melanocytes and sensory nerve endings are derived from the neural crest, which further underscores the evolutionary complexity and specialization of skin structures. The Hedgehog (Hh) signaling pathway plays a critical role in promoting keratinocyte proliferation and hair follicle development, while its dysregulation has been implicated in disorders such as basal cell carcinoma. Understanding these evolutionary mechanisms is vital for comprehending normal skin function and the pathological changes associated with aging [21].

## 3. Skin Aging

### 3.1. Brief Review of Skin Aging

Skin aging is a multifaceted process, which is the manifestation of the progressive and accumulated physiological changes that occur over a period [22,23]. The skin exhibits age-related changes like an increase in skin laxity, impaired wound healing, wrinkles, dryness or xerosis, irregular pigmentation, and pruritis. All the structures of skin exhibit different characteristic physiological changes during the aging process. As the skin starts to age, the proliferation capacity of the keratinocytes reduces due to a reduction in the mitotic capacity of the keratinocytes. This results in reduced epidermal thickness over the years, as a person ages [24]. Also, there is an increase in the number of senescent melanocytes, which also results in loss of hair color with age [25,26]. Photoaging also results in a reduction in the number of Langerhans cells. With age, the epidermis also exhibits dryness due to a compromised skin barrier. With age, the natural moisturizing factor reduces, changes in the lipidic composition of the stratum corneum occur, and a reduction in lipidic enzymes takes place, which results in skin dryness. Sebum production reduces with age, resulting in overall dryness and compromised skin barrier [27]. With age, the structure of dermo-epidermal junctions (DEJ) becomes impaired and the amount of collagen fibers and laminin-332 is reduced. DEJ becomes flattened and the surface area is reduced with age. The fibroblasts change with an increase in the levels of pro-inflammatory cytokines, MMPs, and pro-oxidants. Elevated MMP levels and the atrophy of ECM are the characteristic features of an aging dermis. The atrophy of ECM proteins results in the collapse of the dermis layer. Along with an increase in MMP levels, the atrophy of the dermis is increased. The atrophy of the hypodermis is also seen with aging, which further exacerbates the appearance of wrinkles, reduced thermoregulation, increased skin sagging, and reduced resistance to trauma.

### 3.2. Mechanisms

The chronological aging of skin tissue is genetically programmed and inevitable, resulting from the cumulative impact of various internal and external factors (Figure 2). These sequential changes lead to cellular damage and ultimately compromise the skin’s structural integrity. Understanding the factors driving these changes is essential for enabling skin organoids to accurately simulate the aging process in skin [28].

#### 3.2.1. Intrinsic Aging

Intrinsic aging is a natural, biologically programmed process driven by genetic and molecular changes within the skin over time [29,30]. Key genetic factors, such as telomere length and the function of DNA repair mechanisms, play critical roles [31]. Telomeres, the protective caps at the ends of chromosomes, shorten with each cell division, eventually leading to cellular senescence when they become critically short [32,33]. Variants in genes involved in DNA repair, like Werner syndrome ATP-dependent helicase (WRN), and superoxide dismutase 2 (SOD2) [34], can influence the rate of aging by affecting the cellular response to oxidative stress and DNA damage [35]. A hallmark of intrinsic aging is the accumulation of senescent cells that have permanently exited the cell cycle. Although senescence serves as a protective mechanism against cancer, the persistence of these cells contributes to tissue dysfunction. Senescent cells secrete pro-inflammatory cytokines, growth factors, and proteases, collectively known as the senescence-associated secretory phenotype (SASP), which disrupts normal tissue function, leading to the breakdown of collagen, increased elastin degradation, and impaired skin repair [36]. Mitochondrial function also declines in the aging skin cells, resulting in reduced energy production and increased production of reactive oxygen species (ROS) [37,38], which cause oxidative damage to cellular components, including DNA, proteins, and lipids. Mitochondrial DNA, being particularly susceptible to damage, accumulates mutations over time, further impairing mitochondrial efficiency [39]. This decline in mitochondrial function contributes to the loss of cellular function and the onset of skin aging, which is manifested by thinning and sagging skin [40].

#### 3.2.2. Extrinsic Aging

Extrinsic aging, or environmental aging, is primarily driven by external factors such as UV radiation and pollution. UV radiation is a major contributor to photoaging that induces DNA damage, particularly in the p53 gene, and increases ROS production. This leads to the collagen breakdown, and elastin disorganization, resulting in the characteristic leathery texture of photoaged skin [41]. Additionally, pollutants like particulate matter (PM) and polycyclic aromatic hydrocarbons (PAHs) can accelerate the aging process by exacerbating oxidative stress and inflammation. The combination of UV exposure and pollutants further damages DNA and cellular structures, worsening skin aging. Lifestyle choices, including diet and smoking, significantly impact extrinsic aging. A diet rich in antioxidants supports skin health by neutralizing ROS and promoting collagen synthesis, whereas poor nutrition can exacerbate oxidative damage and impair skin elasticity [42]. Smoking, one of the most potent accelerators of skin aging, reduces blood flow and oxygen delivery to the skin, impairs cellular repair, and promotes ROS and matrix metalloproteinase (MMP) production [43,44]. The MMPs further degrade collagen and elastin, contributing to the formation of deep wrinkles and loss of skin firmness [45].

## 4. Development and Characteristics of Skin Organoids

In a mammal’s life, stem cells form its foundation. Stem cells are the main component for the development of organoids. Skin organoids are advanced in vitro models developed primarily using adult stem cells (ASCs), pluripotent stem cells (PSCs), and tumor cells, which replicate the complex structure and function of human skin [46,47]. ASCs, such as melanocyte and epidermal stem cells, are critical for tissue repair and regeneration, enabling the formation of organoids that closely resemble natural skin, including structures like hair follicles [48]. In a study, the epidermis and dermis from the newborn mouse skin were utilized for the development of skin organoids in vitro. The epidermal cells and dermal cells were combined in a proportion of 1:9, and after 10 days of culturing, skin organoids were formed. During this duration, cells underwent different stages of developing hair follicles, dissociation, aggregation, polarized cyst formation, the merger of cysts, planar skin formation, and eventually the skin had hairs. Further, this skin organoid, when transplanted onto the back of mice, resulted in the development of hairs on mice skin. These findings are important for utilizing this approach for clinically inducing hair growth in patients with baldness and hair loss [49]. PSCs, including the induced pluripotent stem cells (iPSCs), can differentiate into any skin cell type, making them ideal seed cells for creating organoids that mimic fetal skin development and can be used in wound-healing research. Skin organoids with all the appendages, like adipocytes, hair follicles, and sebaceous glands, have been developed from mouse PSCs. In another study, a skin organoid was constructed using human PSCs, i.e., human embryonic stem cells (hESCs). The culture was exposed to BMP and a TGFβ inhibitor. After 75 days, hair follicles started appearing. The hair follicles developed were similar to the mammalian hair follicles. The skin organoids closely mimic the structure and function of human skin, supporting their potential clinical application in treating wounds and burn injuries [50]. Additionally, in vitro models (based on cell lines) and xenograft models (cell-based or patient-derived) all pose a risk of undue adverse events along with huge economic implications. Tumor-derived organoids, such as those from melanoma and keloid tissues, provide valuable models for studying disease mechanisms and screening potential therapies. In a study, fresh tumor tissues were separated by treating them with enzymes and mincing. Once separated, the tissues were segregated according to their size and the tissues in the size range of 40–100 µm were taken for the development of organoids. The selected tissue samples were suspended in suitable media containing type I collagen. After passing the collagen media with the sample through a microfluidic device, the mixture was cultured in Dulbecco’s modified Eagle (DMEM) media, containing a mixture of FBS, glucose, pen/strep, and sodium pyruvate, to construct a spheroid organoid containing multiple cells. The cytokine profiling and immunophenotyping of the melanoma organoid cells by flow cytometry and immunofluorescence staining were found to have preserved the autoimmune cells. Further, this melanoma organoid was used for evaluating drug effectiveness on melanoma [51]. Thus, organoids from tumor tissue seem to be a potentially suitable approach for screening drug candidates and elucidating disease biology. Therefore, skin organoids serve as powerful tools for studying skin biology, pathological conditions, and developing new treatments.

The generation of skin organoids is increasingly reliant on advanced 3D culture techniques, which provide a more physiologically relevant environment compared to traditional 2D cultures. In a 3D setting, cells can interact in a manner that mimics their natural organization within tissues [52,53]. These interactions are essential for the development of complex structures like hair follicles, sweat glands, and the layered architecture of the skin. Biomaterials and scaffolding are critical to the structural development of skin organoids. Materials like Matrigel and collagen scaffolds provide the necessary framework for cells to organize into three-dimensional structures, thereby facilitating the formation of complex skin layers and appendages. These scaffolds not only maintain the shape of developing organoids but also provide essential biochemical cues that guide cell differentiation and tissue formation. For instance, co-culturing sebocytes with keratinocytes on a scaffold offers a suitable model for studying drug responses in a controlled environment [54]. One of the defining features of skin organoids is their ability to replicate the multicellularity of natural skin. Skin organoids consist of various cell types, including keratinocytes, melanocytes, and fibroblasts, which are essential for forming the distinct layers and structures of the skin. This multicellularity ensures that skin organoids exhibit structural features closely resembling native skin, such as the stratified epidermis, dermal–epidermal junctions, and subcutaneous fat layers, which are essential for studying skin diseases and testing novel therapies. Beyond structural mimicry, skin organoids also demonstrate the critical functional characteristics of native skin, including barrier function and the development of skin appendages like hair follicles and sweat glands. These organoids have been shown to replicate essential features of human skin, including the ability to undergo regular epidermal turnover and maintain a protective barrier against environmental damage. The inclusion of functional appendages in organoid models enhances their utility for studying skin physiology, pathology, and the efficacy of potential treatments [55]. The development of skin organoids from different sources of stem cells is graphically depicted in Figure 3.

## 5. Simulation of Skin-Aging Models Using Organoids Models

### 5.1. Cultivation for Intrinsic Aging

Intrinsic skin aging in human skin is a gradual process, marked by a steady decline in cellular function, structural integrity, and regenerative capacity. To model these slow-developing changes, skin organoids are cultured over prolonged periods under conditions that mimic the aging microenvironment [56]. Culture protocols have been developed to maintain these organoids for extended periods, during which they accumulate characteristics typical of aged skin, such as the decreased proliferation of basal keratinocytes, the thinning of the epidermis, and the disorganization of collagen fibers within the dermis. For instance, culture media formulated with reduced oxygen levels and altered nutrient compositions can replicate the in vivo conditions of intrinsic aging, thus providing a dynamic platform for studying the temporal progression of aging at cellular and tissue levels. Furthermore, genetic engineering techniques, particularly CRISPR-Cas9, have been used to introduce mutations associated with premature aging syndromes or key regulatory pathways in skin organoids in order to dissect the molecular mechanisms of skin aging [57]. Knocking out the WRN gene leads to accelerated telomere shortening and genomic instability, which are hallmarks of aged skin. Similarly, manipulating genes like SOD2 or TP53 within organoids can help elucidate the roles of oxidative stress and cell cycle regulation in aging. These genetically modified organoids exhibit phenotypes of accelerated aging, such as increased cellular senescence, impaired barrier function, and reduced dermal elasticity, thus allowing for the identification of the genetic underpinnings of skin aging and the development of potential therapeutic interventions.

### 5.2. Simulation of Extrinsic Aging Through Environmental and Mechanical Exposures

Extrinsic aging, particularly photoaging, is largely driven by chronic exposure to UV radiation, which accelerates skin aging by inducing DNA damage, oxidative stress, and ECM degradation [58]. Photoaging is modeled in skin organoids via controlled exposure to repeated doses of UVB and UVA radiation, which mimics the cumulative effects of sunlight on the skin, resulting in the characteristic signs of photoaging such as increased MMP activity, collagen degradation, elastin fragmentation, and the formation of deep wrinkles [59]. The integration of advanced imaging techniques in the organoid models, such as multiphoton microscopy, can help visualize real-time changes in the dermal architecture and track the onset and progression of photoaging at a cellular level. Moreover, these models can be used to test the efficacy of photoprotective agents, sunscreens, and antioxidants that aim to mitigate or reverse the damage caused by UV exposure. Pollution, particularly airborne PM, is increasingly recognized as a significant factor in extrinsic skin aging. Organoids exposed to pollutants such as PM 2.5 exhibit signs of environmentally stressed skin, such as increased ROS production, the disruption of the stratum corneum, and impaired barrier function [60]. By simulating these conditions in vitro, researchers can study the molecular mechanisms by which pollutants accelerate skin aging, such as the upregulation of inflammatory pathways and the breakdown of tight junction proteins [61]. Microfluidic devices that simulate continuous exposure to pollutants can be integrated into the organoid cultures to provide a more realistic and dynamic environment for studying the chronic effects of pollution on skin health. Taken together, skin organoids can significantly aid in developing anti-pollution skincare products and therapies aimed at restoring barrier function and preventing pollutant-induced aging.

### 5.3. Induction of Cellular Senescence in Skin Organoids

Senescence can be induced in skin organoids through a variety of means, each replicating different aspects of the aging process. Chemotherapeutic agents such as doxorubicin can trigger DNA damage and cellular senescence. The senescent cells within skin organoids can be detected using markers such as cyclin-dependent kinase inhibitor 2A [62], senescence-associated βeta-galactosidase (SA-β-gal) [63], and through changes in cell morphology. These models offer a robust platform for studying the secretion of SASP factors, which are known to exacerbate tissue degradation and inflammation, and exploring possible therapies that can selectively clear senescent cells from aging tissues. Furthermore, incorporating immune cells into senescent skin organoids can model the complex interactions between senescent cells and the immune system, particularly in the context of “inflammaging”, a chronic, low-level inflammation that accelerates aging and contributes to age-related diseases [64,65]. Co-culturing skin organoids with immune cells such as macrophages, T cells, and dendritic cells provides the opportunity to observe the behavior of immune cells in response to senescent cells, including the recruitment of inflammatory mediators and the potential clearance of senescent cells. These advanced models can provide critical insights into the pathways through which chronic inflammation disrupts tissue homeostasis, increases susceptibility to infections, and impairs wound healing [66]. Additionally, they serve as a valuable platform for testing anti-inflammatory therapies and immunomodulators aimed at reducing inflammaging in the skin while preserving immune defenses.

### 5.4. Incorporating Complex Tissue Structures and Systemic Interactions

A key challenge in modeling skin aging is replicating the decline in vascular function that accompanies aging [67]. Recent advances in bioengineering have enabled the successful vascularization of skin organoids by incorporating endothelial cells that form microvascular networks within the organoid structure [68]. These vascularized organoids can accurately replicate the physiological state of human skin, including the delivery of oxygen and nutrients, and the removal of metabolic waste. Vascular aging within these models can be induced through the application of pro-atherogenic factors or by simulating chronic low-grade inflammation, both of which contribute to the vascular thickening and dysfunction seen in aged skin [69]. These models are invaluable for studying the interplay between vascular health and skin aging, particularly in the context of age-related diseases like diabetes, where microvascular dysfunction significantly impacts skin integrity and repair. As mentioned above, the integration of immune cells into skin organoids represents a significant advancement in the ability to study inflammaging that accelerates aging and contributes to the development of age-related pathologies. By incorporating immune cells, researchers can model the inflammatory microenvironment of aged skin, and observe the interaction of immune cells with aging keratinocytes, fibroblasts, and endothelial cells [70].

## 6. Applications of Skin-Aging Organoids

The major applications of the skin organoids are represented in Figure 4. Skin aging organoids can be used to model the progression of age-related skin cancers, such as melanoma, basal cell carcinoma (BCC), and squamous cell carcinoma (SCC) [71]. As the skin ages, the cumulative effects of DNA damage, particularly from UV radiation, increase the risk of carcinogenesis. Gene encoding, the tumor protein p53 (TP53), B-Raf proto-oncogene, serine/threonine kinase (BRAF), and Patched 1 (PTCH1) genes are frequently mutated in skin cancers. Therefore, skin organoids harboring these mutations allow researchers to study the early stages of tumor development, the role of the aging microenvironment in cancer progression, and the response of cancerous cells to various treatments [72]. Additionally, they provide a platform for testing new anti-cancer therapies, including targeted drugs and immunotherapies, under conditions that closely resemble aged human skin. The age-related decline in skin regenerative capacity often leads to chronic wounds and ulcers, particularly in individuals with underlying conditions such as diabetes or vascular insufficiency. Skin-aging organoids offer a robust model for studying the impaired wound-healing processes associated with aging. By replicating the reduced proliferation of keratinocytes, altered ECM composition, and diminished angiogenesis seen in aged skin, these organoids can be used to investigate the molecular pathways that hinder wound closure and tissue repair, screen for compounds that enhance wound healing in aged skin, and develop therapies aimed at improving outcomes in elderly patients with chronic wounds [73]. Aging skin is also more prone to fibrosis and scarring due to the increased activity of fibroblasts and the excessive deposition of ECM components [74,75]. Skin-aging organoids serve as an excellent model for studying the mechanisms of fibrosis and the development of hypertrophic scars or keloids in aged skin. Furthermore, these organoids can be manipulated to overexpress fibrotic markers such as TGF-β, and form tissues that mimics the pathological scarring seen in aging individuals. Such models are crucial for identifying potential anti-fibrotic agents and assessing the impact of aging on the balance between tissue repair and fibrosis [76,77].

Skin organoids offer a controlled environment for evaluating the efficacy and safety of anti-aging compounds [78]. These models have been extensively used to test compounds that promote collagen synthesis, reduce MMP activity, or enhance cell proliferation, assessing their effects on reversing age-related skin damage. For instance, retinoic acid, a well-known anti-aging compound, has been tested on photoaged skin organoids, where it increased keratinocyte proliferation, improved epidermal thickness, and reduced ROS levels in organoid models of photoaging, thereby demonstrating its potential to mitigate the effects of extrinsic aging. Skin organoids thus provide a high-throughput platform for testing a wide range of compounds, from antioxidants to growth factors, aimed at restoring youthful skin function. Senolytics, a class of drugs designed to selectively eliminate senescent cells, are increasingly being recognized for their potential in treating skin aging [79] and have been tested in skin organoids, serving as valuable tools for testing these drugs in a controlled manner [80]. For example, the Solidago virgaurea extract successfully removed senescent melanocytes from aging skin organoids, improved epidermal thickness, and reduced the amount of inflammation markers [81]. Anti-glycation agents represent another promising therapeutic avenue, targeting advanced glycation end products (AGEs) that accumulate in the skin’s ECM and cross-link collagen and elastin fibers, reducing skin elasticity and promoting aging [82]. These novel therapeutic agents have shown promising results in preventing AGE accumulation, partly reversing glycation-induced damage, improving collagen organization, and restoring ECM integrity in skin organoids.

Skin-aging organoids provide a unique platform for dissecting the cellular and molecular pathways involved in skin aging. By overexpressing or knocking down specific genes in these organoids, the roles of oxidative stress, DNA repair, and cellular senescence-related pathways in aging can be dissected in greater detail [83]. Additionally, advanced imaging and omics technologies can be applied to these organoids to monitor changes in gene expression, protein production, and metabolic activity over time, offering molecular insights into the aging process at a granular level. The microenvironment of aged skin is characterized by alterations in the ECM, increased oxidative stress, and chronic inflammation, all of which contribute to the decline in skin function [84]. Skin-aging organoids can be engineered to replicate these microenvironmental changes, and study their impact on cellular behavior and tissue integrity. For example, these organoids can be used to determine how changes in ECM stiffness affect fibroblast activity, or the impact of chronic exposure to pro-inflammatory cytokines on keratinocyte differentiation and barrier function.

The cosmetic and skincare industries are increasingly turning to organoids with human-skin-like structures to develop and test new products aimed at combating the signs of aging [85]. These organoids provide a more precise and ethically responsible alternative to animal testing. In addition, peptides, antioxidants, hyaluronic acid, and other actives can be tested on aged skin organoids to determine their effectiveness in reducing wrinkles, enhancing hydration, and improving overall skin texture [86]. The use of organoids in skincare research also allows for the assessment of product safety, ensuring that new formulations do not cause irritation or adverse effects in aged skin [87]. Furthermore, skin-aging organoids can also be used to evaluate the effectiveness of cosmetic treatments, such as laser therapies, microneedling, and chemical peels, along with their mechanisms of action at a cellular level, and potential long-term benefits. For example, organoids can be used to assess the impact of laser treatments on collagen production and ECM remodeling, and optimize the treatment protocols for aging skin [88]. This application is particularly valuable for developing non-invasive cosmetic procedures that offer visible improvements in skin appearance without the need for surgery.

## 7. Future, Challenges, and Limitations

### 7.1. Future

The use of patient-specific cells to create skin organoids marks a significant advancement in personalized medicine. The iPSCs derived from a patient’s own skin cells can generate organoids that accurately reflect the genetic and epigenetic characteristics of the individual’s skin [89,90]. These organoids can model intrinsic aging processes, such as cellular senescence, collagen production decline, and altered responses to oxidative stress, thereby providing a personalized view of aging. Moreover, exposing these organoids to environmental stressors like UV radiation or pollution would allow researchers to study the differences in skin response to these extrinsic aging factors among individuals [91]. This approach is particularly valuable for researching rare genetic aging disorders and for tailoring therapeutic strategies to an individual’s genetic profiles. The ultimate goal of utilizing personalized skin organoid models is to develop individualized anti-aging therapies. Given the genetic diversity and variety of factors contributing to skin aging, personalized organoids offer a unique platform for targeted drug screening [92]. For instance, a patient with a genetic predisposition affecting collagen production might benefit from therapies designed to enhance ECM synthesis, while another patient with heightened oxidative stress could be more effectively treated with antioxidant therapies. Personalized organoids allow for the testing of such treatments in a patient-specific context, ensuring that the therapies are both effective and safe before clinical application [93]. This approach has the potential to revolutionize anti-aging therapy by making it more effective, safer, and tailored to individual needs.

Artificial intelligence (AI) and machine learning (ML) are increasingly being integrated into skin organoid research, offering new avenues for analyzing complex datasets and predicting the effects of anti-aging interventions [94,95]. AI and ML algorithms can process vast amounts of data generated from high-throughput screening (HTS) and imaging studies and identify subtle patterns and correlations that might otherwise be missed [96]. In skin-aging research, AI-driven approaches are highly effective for modeling the aging process over time. Deep learning models can simulate aging progression by analyzing cellular morphology, molecular markers, and ECM composition in real-time. AI also enhances the reproducibility and scalability of organoid maintenance, automating monitoring processes crucial for anti-aging studies. Additionally, ML algorithms streamline drug discovery by identifying potential anti-aging compounds and predicting their efficacy. By training on extensive datasets of tested compounds, AI systems can forecast the long-term impact of new treatments on skin organoid models [97,98]. The integrations of AI and ML can facilitate the discovery of new therapeutic targets and optimize the development of anti-aging compounds by predicting their long-term efficacy based on early data.

### 7.2. Challenges and Limitations

Despite the significant advancements in skin organoid technology, there are challenges pertaining to the reproducibility and scalability of these models. The complexity of organoid development, from cell sourcing to precise culture conditions, can lead to variability in results across different experiments and laboratories. Even small differences in growth factors, biomaterials, or differentiation timing can result in significant variations in organoid morphology and function, making it difficult to establish consistent protocols for skin-aging research. Additionally, the current methods for generating skin organoids are labor-intensive and require meticulous handling, thus limiting their scalability for drug screening or clinical applications. Efforts to automate organoid production using bioprinting or high-throughput systems are underway, but face significant technical challenges that need to be addressed to make these models more practical and widespread. Such advancements will create even more relevant models, allowing researchers to study the intricate interactions between different skin components and their collective impact on aging.

The advances in skin organoid technology have also brought forth ethical considerations [99]. The creation of organoids that closely mimic human tissues raises questions regarding the extent to which these models resemble actual human organs and the ethical boundaries that should govern their use. For instance, skin organoids incorporating sensory neurons or immune cells might raise concerns about the potential for these models to experience stimuli in ways that could be considered too close to human experiences. Additionally, the use of patient-derived cells for personalized organoid creation raises important issues regarding consent, privacy, and the ownership of biological material. Establishing clear ethical guidelines will be crucial as personalized organoid technology advances to clinical translation, ensuring that it is used responsibly and that patient rights are protected.

## 8. Conclusions

Skin organoids have emerged as a transformative tool in aging research, providing a sophisticated in vitro model to study the complex biological processes of skin aging. By accurately replicating the structure and function of human skin, these organoids facilitate the study of genetic predispositions, environmental factors, and oxidative stress involved in skin aging. Advances in stem cell technology, 3D bioprinting, and gene editing are set to further enhance the precision and relevance of these models, paving the way for personalized anti-aging therapies tailored to individual genetic profiles. However, significant challenges remain, particularly regarding reproducibility, scalability, and cost, which must be addressed to fully realize the potential of skin organoids in both research and clinical applications. Continued interdisciplinary collaboration and technological innovation will be crucial in overcoming these barriers, ultimately advancing the field of dermatology and improving patient outcomes.

## Figures and Tables

**Figure 1 biomolecules-14-01436-f001:**
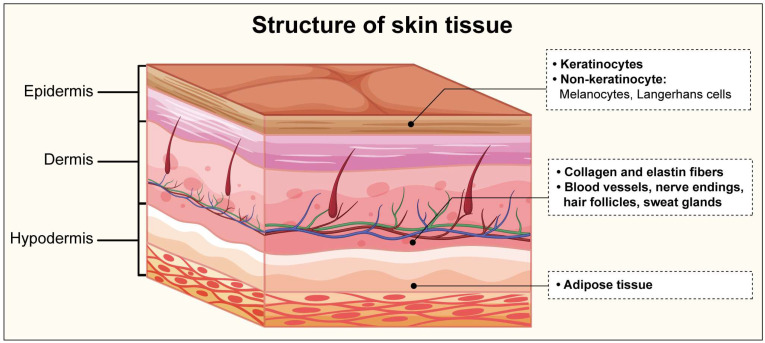
Structure of skin tissue.

**Figure 2 biomolecules-14-01436-f002:**
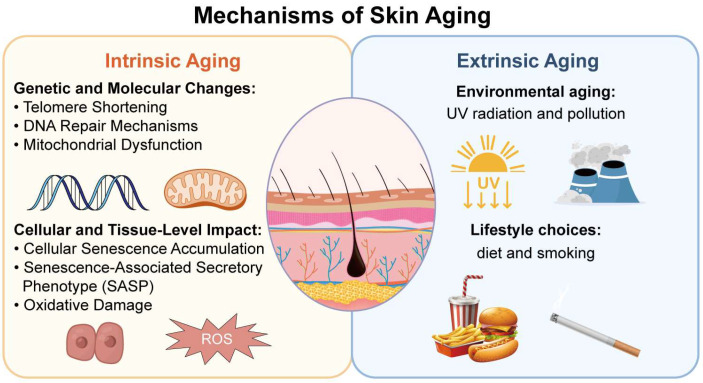
Brief mechanism of skin aging.

**Figure 3 biomolecules-14-01436-f003:**
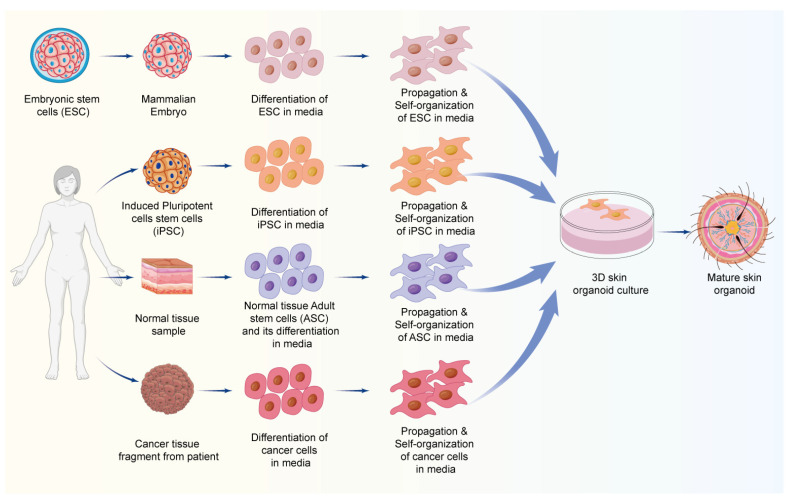
A schematic representation of the general process of skin organoid development from various sources of stem cells, viz., embryonic stem cells (ESCs), induced pluripotent cells (iPSCs), adult stem cells (ASCs), and cancer tissue cells in a culture media.

**Figure 4 biomolecules-14-01436-f004:**
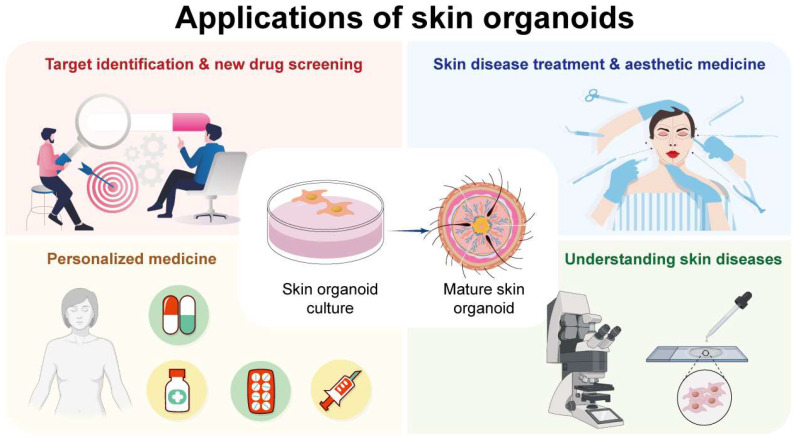
Applications of skin organoids: from culture to drug screening, aging research, and personalized medicine.

## Data Availability

Not applicable.

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
