# Peer review of "Organoids as Tools for Investigating Skin Aging: Mechanisms, Applications, and Insights"

_biomolecules, 2024, doi:10.3390/biom14111436_

Round 1

Reviewer 1 Report

Comments and Suggestions for Authors

The review is very interesting and well written. However, some improvements could be done.

- In the second paragraph in section 2 (A brief overview of the ski), the authors describe the skin perfectly, but no reference is cited. Please, include a reference for that information.

- More references are needed throughout the review.

- More illustrative figures would help the reader to better understand all the details and explanations

Author Response

Dear Reviewer,

Thank you very much for your positive feedback and constructive suggestions regarding our review. We greatly appreciate the time and effort you have taken to improve the quality of our manuscript. Below, we have addressed each of your comments in detail:

Question 1: “In the second paragraph in section 2 (A brief overview of the ski), the authors describe the skin perfectly, but no reference is cited. Please, include a reference for that information.”

Our response: We sincerely apologize for the oversight and have now included a relevant reference to support the description of skin in section 2. We believe this addition enhances the credibility of the content.

Question 2: “More references are needed throughout the review.

Our response: We carefully reviewed the entire manuscript and added more citations where necessary, particularly in sections that were previously under-referenced. These additional references are drawn from recent and relevant studies to ensure the review is comprehensive and well-supported by existing literature.

Question 3: “More illustrative figures would help the reader to better understand all the details and explanations.”

Our response: We highly agree that additional figures would help clarify key points in the review. To address this, we have included new illustrative figures that better visualize the processes and mechanisms discussed, including the development of skin organoids and their applications. We believe these additions will enhance the reader's understanding.

For a comprehensive view of the revisions, we kindly invite you to refer to the newly revised manuscript. The red highlights indicate the modifications we made to the manuscript. Once again, we express our gratitude for your thoughtful review and constructive feedback. Your input has been instrumental in elevating the scholarly merit of our manuscript. We are confident that these improvements have strengthened the scientific value of our research. If there are any additional concerns or suggestions, we are eager to consider them further. Your further guidance will undoubtedly aid us in presenting the most comprehensive and accurate study possible.

Best regards,

Jun Li

Reviewer 2 Report

Comments and Suggestions for Authors

The present review describes possible approaches to study skin aging using skin organoids. The main components of the skin and its structure, mechanisms of skin aging and its causes are described. However, the section "Development and characteristics of skin organoids" that should describe skin organoids is quite short, approaches to skin modeling are described quite poor. It would be better to improve this section by adding more detailed descriptions of existing models of skin organoids and the results obtained using them. In addition, many of the statements in the review are not supported by references to literature sources, and the references list itself is mainly represented by reviews, rather than research papers.

Page 4 - the headline "Multicellularity and Structural Resurrection to Native Skin" is not highlighted.

Author Response

Dear Reviewer,

We would like to extend our heartfelt thanks for your thorough review and active feedback. Your expertise and guidance have undoubtedly contributed to the advancement of our research. We remain committed to producing a high-quality research article that contributes significantly to the field. Below, we provide a point-by-point response:

Question 1: “Expansion of the section on "Development and Characteristics of Skin Organoids

Our response: We highly agree with your suggestion that this section could benefit from further elaboration. We have expanded this section by including a more detailed discussion of existing models of skin organoids, including both traditional and cutting-edge approaches. We have also added descriptions of the specific results obtained using these models to offer more comprehensive insights into their applications in studying skin aging.

Question 2: “More references to research papers rather than reviews

Our response: We have revisited the references throughout the manuscript and replaced many of the review citations with primary research papers to provide a stronger foundation for the statements made. This ensures that the review is more grounded in original research and reflects the latest advancements in the field. Additionally, we have incorporated numerous (>20) cutting-edge references to ensure the inclusion of the latest research advancements.

Question 3: “Headline on Page 4 ("Multicellularity and Structural Resurrection to Native Skin

Our response: We sincerely apologize for the oversight in not removing some outline items from the manuscript. We regret any inconvenience this may have caused and appreciate your understanding. We have now carefully reviewed the manuscript to ensure these items have been properly removed.

For a comprehensive view of the revisions, we kindly invite you to refer to the newly revised manuscript. The red highlights indicate the modifications we made to the manuscript. We believe these changes address your concerns and significantly improve the clarity and depth of the review. Thank you again for your valuable insights, which have helped to strengthen our manuscript. We look forward to your further comments.

Best regards,

Jun Li

Reviewer 3 Report

Comments and Suggestions for Authors

Abstract: Authors mention "3D bioprinting, and AI-driven" and there is no information rofund during the manuscript

1. Introduction: there a lot of information missing here of tradition models. What about spheroids, 3D sophisticated skin models (they are referred later)

2. Brief of skin: a representative graph is needed. Please, introduce a subsection with the information of "evolutionarily conserved signaling pathways, including Wnt/β-catenin, fibroblast growth factors (FGF)," and porovide more signaling pathways. Missing the concetp of organoid and how can be formed. biomateriales used?

3. Mechanisms of Skin Aging. A represenative graph is needed. Please provide the meaning of differente abreviations (for example, WRN and SOD2)

3.3. Molecular and cellular changes during skin aging. The title does not correspond with the content.

4. Development and characteristics of skin organoids. More information are needed to describe the different skin organoids in literature. whta are teh advantages vs traditional skin models? what is "Multicellularity and Structural Resemblance to Native Skin."?

Figure 1: where are (PSCs)? Please, provide the origin of the figures

5. Creating Skin Aging Models Using Organoids. Please porvide a represenative graph of this section. 5.1. Long-Term Cultivation for Intrinsic Aging: what time is prolonged? missing information  and more details of different cases ofund in literature

5.2. Simulating Extrinsic Aging Through Environmental and Mechanical Exposure. Mor einformation is needed. What is PM2.5? In general, many statements without refs (along the manuscript)

5.3. Inducing cellular senescence in skin organoids. Senescence can be triggered by other means Please, explain that. what are the abbrevations of p16^INK4A^, SA-β-gal,?

5.4. Incorporating complex tissue structures and systemic interactions. Statements without refs

6. Applications of skin aging organoids.Statements without refs

Figure 2 is very poor

Comments on the Quality of English Language

English must be improved

Author Response

Dear Reviewer,

We would like to express our sincere gratitude for your valuable feedback on our manuscript. Your insightful comments and suggestions have been immensely helpful in shaping our research and improving the quality of our work. We truly appreciate the time and effort you have dedicated to reviewing our article. We are committed to addressing the specific areas you highlighted and making necessary improvements to strengthen the overall coherence and clarity of the paper. Your recommendations have provided us with a valuable roadmap for refining our findings and enhancing the overall impact of our study. As per your expert recommendations, we have carefully incorporated the following revisions:

Question: “Abstract: You pointed out the mention of "3D bioprinting and AI-driven" technologies in the abstract without sufficient elaboration in the manuscript.

Our response: We highly appreciate your observation regarding the mention of "3D bioprinting and AI-driven" technologies in the abstract without sufficient elaboration in the manuscript. We understand the importance of maintaining consistency between the abstract and the main content, and we sincerely apologize for the oversight. In the revised manuscript, we included a section that elaborates on recent advancements in 3D bioprinting for skin organoid development, highlighting how these techniques enable precise control over the spatial arrangement of cells and extracellular matrix components, which is crucial for replicating the complex structure of skin. Besides, we added more content to explain the role of AI in optimizing organoid culture conditions, automating data analysis, and predicting therapeutic outcomes. To support these new sections, we also included references to recent studies that showcase the integration of 3D bioprinting and AI-driven technologies in the field of skin organoid research. This will not only strengthen the manuscript but also provide readers with a comprehensive overview of these cutting-edge approaches.

Question 1: “Introduction: there a lot of information missing here of tradition models. What about spheroids, 3D sophisticated skin models (they are referred later)

Our response: Thank you for your valuable feedback. We acknowledge the need to further elaborate on traditional models, specifically spheroids and advanced 3D skin models, to provide a more complete background for the discussion of skin organoids. We have updated the manuscript to address these gaps.

Question 2-1: “Brief of skin: a representative graph is needed

Our response: We highly agree that a representative graphic is essential for this section, and we include one to visually describe the main components of the skin.

Question 2-2: “Please, introduce a subsection with the information of "evolutionarily conserved signaling pathways, including Wnt/β-catenin, fibroblast growth factors (FGF)," and porovide more signaling pathways.

Our response: We added a subsection focusing on evolutionary conserved signaling pathways, as you suggested, and expand on other relevant pathways.

Question 3-1: “Mechanisms of Skin Aging. A represenative graph is needed.

Our response: We completely agree with your perspective, and we acknowledge that adding an image at this point in the manuscript would be highly beneficial. We have accordingly added relevant and informative figure to this section to better illustrate the content, ensuring a clearer and more engaging presentation.

Question 3-2: “Please provide the meaning of differente abreviations (for example, WRN and SOD2)” AND “What is PM2.5?” AND “Please, explain that. what are the abbrevations of p16^INK4A^, SA-β-gal,?”.

Our response: We apologize for the oversight in not providing clear explanations for the abbreviations. We understand that this may have caused some confusion, and we appreciate your suggestion to address this issue. In response, we have revised the manuscript to ensure that all abbreviations are properly defined upon their first mention. We hope this adjustment will improve the clarity and readability of the text.

Question 3-3. “Molecular and cellular changes during skin aging. The title does not correspond with the content.” 

Our response: We sincerely appreciate your insightful feedback, and we fully agree with your observation regarding the lack of logical flow in the section you pointed out. We acknowledge that the content of this segment was repetitive and overlapped with previous parts of the manuscript. To address this, we have carefully revised the manuscript by removing the redundant section and integrating its relevant information into other parts of the text. These changes have been made to enhance the coherence and logical progression of the manuscript, ensuring a more seamless and structured narrative. We believe that this revision will improve the clarity and readability of the manuscript, and we are grateful for your valuable input in helping us achieve this.

Question 4-1: “Development and characteristics of skin organoids. More information are needed to describe the different skin organoids in literature. whta are teh advantages vs traditional skin models?”.

Our response: We highly appreciate your suggestion for further elaboration in this section. We provided a more detailed review of the various types of skin organoids described in the literature, their advantages over traditional models, and how they better replicate skin biology.

Question 4-2: “what is "Multicellularity and Structural Resemblance to Native Skin."?

Our response: We sincerely apologize for the oversight in not removing some outline items from the manuscript. We regret any inconvenience this may have caused and appreciate your understanding. We have now carefully reviewed the manuscript to ensure these items have been properly removed.

Question 4-3: “Figure 1: where are (PSCs)? Please, provide the origin of the figures

Our response: We apologize for the oversight that led to the omission of some information. We have now revised and re-drawn the figure to include the missing details. Thank you for bringing this to our attention, and we hope the updated image provides a clearer and more comprehensive presentation.

Question 5-1: “Creating Skin Aging Models Using Organoids. Please porvide a represenative graph of this section.

Our response: We added a representative graphic for this section to visually summarize the section.

Question 5-2: “5.1. Long-Term Cultivation for Intrinsic Aging: what time is prolonged? missing information and more details of different cases ofund in literature”

Our response: We apologize for the ambiguity caused by our phrasing in English. We have revised this section to improve clarity and ensure that our intended meaning is accurately conveyed. We regret any inconvenience this may have caused and appreciate your understanding. hank you for your attention to this detail.

Question: “Statements without refs” 

Our response: We acknowledge that there are statements lacking proper references in many sections. We sincerely apologize for not including sufficient and appropriate references in the original submission. In response to your suggestion, we have carefully revised the manuscript and added the necessary citations as required. Additionally, we have incorporated numerous (>20) cutting-edge references to ensure the inclusion of the latest research advancements. This ensures that the review is more grounded in original research and reflects the latest advancements in the field.

Question: “Figure 2 is very poor”

Our response: Figure 2 is not sufficiently detailed and does not adequately illustrate the content. We revised this figure to better depict the various applications of skin organoids in aging research.

In summary, we are very grateful for your detailed feedback and will work diligently to address all the issues raised. For a comprehensive view of the revisions, we kindly invite you to refer to the newly revised manuscript. The red highlights indicate the modifications we made to the manuscript. Once again, we express our gratitude for your thoughtful review and constructive feedback. Your input has been instrumental in elevating the scholarly merit of our manuscript. We hope these revisions will meet your expectations and result in a much stronger manuscript. We look forward to your further feedback after revisions.

Best regards,

Jun Li

Round 2

Reviewer 2 Report

Comments and Suggestions for Authors

The authors made changes that significantly improved the quality of this paper and made it more understandable and useful for readers.

Author Response

Dear Reviewer,

Thank you very much for your positive feedback and for recognizing the improvements made to our manuscript. We are grateful for your constructive comments, which greatly contributed to enhancing the clarity and overall quality of the paper. We appreciate your time and effort in reviewing our work and are pleased to know that it now meets your expectations and is more valuable for readers. Your insights were invaluable, and we look forward to any further suggestions you might have.

Best regards,

Jun Li

Reviewer 3 Report

Comments and Suggestions for Authors

Please, explain the origin of all figures (adapted, right permissions etc..)

Comments on the Quality of English Language

English is ok

Author Response

Dear Reviewer,

We sincerely apologize for any confusion regarding the origin of the figures. We would like to assure you that all figures included in our manuscript have been carefully created and reviewed by a dedicated member of our team responsible for illustrations. Some elements have been adapted from previously purchased stock images or sourced from publicly available, copyright-free platforms. We are committed to ensuring that all figures comply with copyright regulations and do not infringe on any third-party rights. We sincerely hope this clarifies any concerns regarding the origin and permissions of the images in our manuscript, and we are open to providing any further information or documentation if needed. Thank you for your attention and understanding. We greatly appreciate your attention to this matter and have updated the manuscript to clearly indicate the origin of each figure to ensure full transparency and compliance.

Best regards,

Jun Li